# Replay Buffer With Local Forgetting for Adaptive Deep Model-Based Reinforcement Learning

## Abstract

One of the key behavioral characteristics used in neuroscience to determine whether the subject of study—be it a rodent or a human—exhibits model-based learning is effective adaptation to local changes in the environment. In reinforcement learning, however, recent work has shown that modern deep model-based reinforcement-learning (MBRL) methods adapt poorly to such changes. An explanation for this mismatch is that MBRL methods are typically designed with sample-efficiency on a single task in mind and the requirements for effective adaptation are substantially higher, both in terms of the learned world model and the planning routine. One particularly challenging requirement is that the learned world model has to be sufficiently accurate throughout relevant parts of the state-space. This is challenging for deep-learning-based world models due to catastrophic forgetting. And while a replay buffer can mitigate the effects of catastrophic forgetting, the traditional first-in-first-out replay buffer precludes effective adaptation due to maintaining stale data. In this work, we show that a conceptually simple variation of this traditional replay buffer is able to overcome this limitation. By removing only samples from the buffer from the local neighbourhood of the newly observed samples, deep world models can be built that maintain their accuracy across the state-space, while also being able to effectively adapt to changes in the reward function. We demonstrate this by applying our replay-buffer variation to a deep version of the classical Dyna method, as well as to recent methods such as PlaNet and DreamerV2, demonstrating that deep model-based methods can adapt effectively as well to local changes in the environment.

## 1 Introduction

Recent work has shown that modern deep MBRL methods adapt poorly to local changes in the environment (Van Seijen et al., 2020; Wan et al., 2022), despite this being a key characteristic of model-based learning in humans and animals (Daw et al., 2011). The analysis by Wan et al. (2022) revealed that there are broadly two causes for this lack of adaptivity: an insufficient world model or insufficient planning. And the former one is an especially challenging one to overcome when deep-learning-based world models are considered. The core of this challenge lies in the fact that adaptivity requires a world model that is accurate across the relevant state-space, as a small change in reward or transition function can change the trajectory of the optimal policy entirely. By contrast, to achieve high single-task sample-efficiency—a common metric used in MBRL research—it is sufficient that the world model is accurate along the current behavior policy.

For deep world models, accuracy across the state-space is hard to achieve and maintain, even with sufficient exploration. The reason is that collected samples are strongly correlated, and, at the final stages of learning, mostly come from states along the trajectory of the optimal policy. Due to catastrophic forgetting, the quality of the predictions further away from this trajectory quickly degrades. A common strategy to counter this is to use a replay buffer. By randomly sampling from a large replay buffer and using these samples to update the world model, the effects of catastrophic forgetting are greatly reduced. However, the traditional first-in-first-out (FIFO) replay buffer has the disadvantage that it hinders effective adaptation, as out-of-date samples interfere with the new data.

To address the challenge of catastrophic forgetting, while also avoiding interference from out-of-date samples, we propose a variation of the traditional FIFO replay buffer. Instead of removing the oldest sample from the replay buffer once the buffer is full, the oldest sample in the *local neighbourhood* of the new sample is removed. This conceptually simple idea naturally leads to a replay buffer whose samples are approximately spread out equally across the space space, while local changes are accounted for quickly. Consequently, updating the deep world model with samples drawn randomly from this replay buffer results in a world model that is approximately accurate across the state-space at each moment in time. We call this replay buffer variation a LOFO (*Lo*cal *Fo*rgetting) replay buffer. One practical challenge to our proposed variation is that a locality-function needs to be learned that determines whether or not a sample from the replay buffers falls within the local neighborhood of a newly observed sample. We train this locality function using contrastive learning (Hadsell et al., 2006; Dosovitskiy et al., 2014; Wu et al., 2018) during the initial stages of learning, after which it is fixed and used as basis for the LOFO replay buffer.

We demonstrate the effectiveness of the LOFO replay buffer by combining it with a deep version of the classical Dyna method and measuring its adaptivity on a variation of the MountainCar task as well as a mini-grid domain, using the same Local Change Adaptation (LoCA) setup as used by Wan et al. (2022). We then test the limits of our approach by applying the same idea to both PlaNet (Hafner et al., 2019b) and DreamerV2 (Hafner et al., 2020), which use world models based on recurrent networks and are intended for continuous-action domains. Experiments with these modified methods on variations of the MuJoCo Reacher domain demonstrate that a LOFO replay buffer can substantially improve adaptivity of more advanced deep MBRL methods as well.

## 2 BACKGROUND: LOCAL CHANGE ADAPTATION (LOCA) SETUP

Inspired by work from neuroscience on detecting model-based behavior in humans and animals (Daw et al., 2011), Van Seijen et al. (2020) proposed the Local Change Adaptation (LoCA) regret as an experimental setup and metric to evaluate model based behavior of RL methods. LoCA regret measures how quickly a method adapts its policy after observing a local environment change and tries to estimate how close a method is to ideal model-based behavior. Wan et al. (2022) improved the LoCA setup by making it simpler, less sensitive to hyperparameters and easily applicable to stochastic environments. It evaluates if an RL method can reach close to optimal performance after observing a local environment change and makes a binary classification of methods that effectively adapt to local changes and those that cannot. In this work we use this improved LoCA setup (Figure 1) for all our evaluations. It consists of two main components, the task configuration and the experiment configuration.

The task configuration considers an environment with two tasks, namely A and B, which differ only in their reward functions. A method's adaptivity is determined by analyzing how effectively it can adapt from task A to task B. A learning environment suited for the LoCA setup is made up of two terminal states, namely T1 and T2. The reward function for both tasks is always zero except when transitioning to ei-

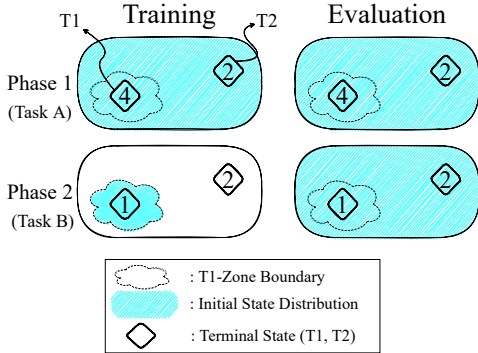

Figure 1: LoCA Setup: During training, in Phase 1, the agent is randomly initialized throughout the state-space and has to solve Task A which has a higher reward in T1 compared to T2. In Phase 2, the agent is initialized within the one-way T1-zone and the reward function changes to Task B with T1's reward changing and having a lower reward than T2, hence changing the optimal policy. During evaluation the agent is initialized randomly throughout the entire state-space in both the phases.

ther of the terminal states. For task A, the agent receives a reward of 4 upon transitioning to T1 and 2 upon transitioning to T2. For task B, however, a transition to T1 yields a reward of 1 (instead of 4 as in task A), and transitioning to T2 results in the same reward as in task A, which is 2. In addition, both of these tasks share the same discount factor $0 < \gamma < 1$. Finally, the transition dynamics for

a local area around T1 is such that it is impossible for an agent to move out, once entered. This is called the *T1-zone*.

The change between the two tasks is local and is only in their reward functions. Specifically, tasks A and B share the same environment dynamics, except that a transition to T1 results in different rewards between the two tasks. In cognitive neuroscience this is also known as reward transfer, or retrospective reward revaluation (Momennejad et al., 2017). While the difference in tasks A and B is only local, their optimal policies are completely different for most states, i.e., the optimal policy for task A tries to reach T1, while for task B, it tries to reach T2 except when the agent is within the T1-zone, where the optimal policy tries to reach T1.

The experiment configuration consists of two different training phases. These two phases differ in the task and initial state distribution. Throughout Phase 1, the environment dynamics are set to that of task A's, and the initial state is drawn uniformly random from the entire state-space. When the first phase finishes, the task changes to task B, and Phase 2 begins. The initial state for the second phase is drawn uniformly random from only states within the T1-zone; hence, the agent's presence is limited to a local area around T1. During each phase, the agent is periodically evaluated for some number of episodes, and the initial states for each evaluation episode is drawn uniformly random over the entire state-space. The average overall return is reported for each evaluation and compared to the average return of the corresponding optimal policy in that phase. Lastly, the agent does not receive any signal when the task changes.

For a method to reach optimal performance in Phase 2, it has to adapt to the new reward function corresponding to task B, i.e., it has to change its policy from pointing towards T1 to pointing towards T2 for most of the state-space. The method has to perform this adaptation to the reward change while only observing states within the local region around T1 where the reward change has occurred. Given sufficient amount of time to train in Phase 1 followed by Phase 2, a method is classified as adaptive if it is able to achieve close to optimal performance in both Phase 1 and Phase 2. If a method achieves near-optimal performance in Phase 1 but not Phase 2, it is classified as a non-adaptive method. And lastly, no assessment is made when a method fails to reach close to optimal performance in Phase 1.

## 3 CHALLENGES IN ADAPTIVE DEEP MBRL

Using the LoCA setup, Wan et al. (2022) observed that current popular deep MBRL methods, such as PlaNet (Hafner et al., 2019b) and DreamerV2 (Hafner et al., 2019a; 2020) fail to adapt to local environment changes. Their analysis revealed that the key reason for their failure to adapt is their inability to build and maintain correct world models when environment changes occur. A small local change in reward or transition function can change the optimal policy entirely. For an MBRL method to adapt to such environment changes and update its policy, it needs to maintain a world model that is sufficiently accurate throughout the relevant parts of the state-space. It is not sufficient to have a world model that is accurate only along the current behavior/optimal policy, as is the case for current methods such as PlaNet and Dreamer, developed for single non-changing task settings.

Current deep MBRL methods that use deep neural networks as function approximators, such as PlaNet and DreamerV2 store their recent experience to a large first-in-first-out (FIFO) replay buffer. In a FIFO replay buffer, when the buffer gets full, as new transition samples are added to the buffer, the oldest samples are removed. Transitions are then sampled from the replay buffer to train the world model, which is then used for planning and updating the agent's policy. Instead of training the model directly from the stream of data that the agent is observing, having a replay buffer and sampling transitions from it to train the deep neural network based model, helps break the strong correlation between the stream of data that the agent is observing. This correlation is hurtful for the i.i.d., assumptions made by the stochastic gradient-descent based optimizers typically used to update the parameters of the neural networks. Replaying past experience also helps mitigate forgetting predictions about states that the agent does not visit frequently.

When a change occurs in the environment, using a large replay buffer results in interference of the old out-of-date data with the new data, resulting in an incorrect world model. For example, in the LoCA setup, when the agent enters Phase 2 with task B, there is a local change in the reward function, specifically the reward corresponding to T1 changes. The agent which is restricted to the

T1-zone in Phase 2 is able to observe this local change. However, the replay buffer from which transitions are sampled to update the model still has lots of transitions from Phase 1 with the stale incorrect reward corresponding to T1 from task A. This interference from the old data results in an incorrect world model, which affects planning and thereby renders the method not adaptive to the local change in Phase 2.

In a FIFO replay buffer, the current samples in the replay buffer are determined by the current behavior policy. Therefore, as time progresses, the old data will all be removed from the buffer. This will happen faster if we make the replay buffer smaller. While it might avoid the interference from old stale data after some time, it would result in catastrophic forgetting of model predictions corresponding to states not recently visited. For example, in the LoCA setup, when the replay buffer is small, after entering Phase 2, the stale transitions around T1 from phase 1 will be removed from the replay buffer over time, stopping the interference problem. This, however, also removes other still relevant and correct transitions from other parts of the state-space, which are now replaced by transitions from just within the T1-zone. A model trained with such a replay buffer forgets the learned model for states outside the T1-zone, which again affects the planning and renders the method not adaptive to the changes in Phase 2.

Therefore, the core challenge in building adaptive deep MBRL methods is to address the interference-forgetting dilemma and maintain world models that are sufficiently accurate throughout the relevant state-space when tasks change, irrespective of the current behavior policy.

## 4 Local Forgetting (LoFo) Replay Buffer For Adaptive Deep MBRL

We propose LoFo (*Local Forgetting*) replay buffer, a conceptually simple variation of the traditional FIFO replay buffer that is able to address the core challenges in building an adaptive deep MBRL method. Instead of removing the oldest samples from the replay buffer once it is full as done in the traditional FIFO replay buffer, in a LoFo replay buffer, the oldest samples in only the *local neighbourhood* of the new samples are removed.

When a change occurs in the environment and the agent observes that change, adding the new samples to the replay buffer and removing the oldest samples from only the local neighbourhood of the new data instead of the oldest samples from the entire replay buffer facilitates removal of the potentially incorrect and stale data sooner. This helps mitigate the interference of the old out-of-date samples with the new samples when tasks change.

Since only the samples in a local neighbourhood of the new samples are removed, old samples in other parts of the state-space including those that have not been visited recently remain in the replay buffer. This leads to a replay buffer whose samples are approximately spread out equally throughout the entire relevant state-space, irrespective of the current behavior policy. Updating the world model with samples drawn from this LoFo replay buffer results in a world model that is approximately accurate throughout the relevant state-space avoiding the problem of catastrophic forgetting of predictions related to states not recently visited.

LoFo replay buffer can therefore address the interference-forgetting dilemma and result in learning a world model that is sufficiently accurate throughout the relevant state-space even when there are environment changes. The current deep MBRL methods can then utilise the correct world models to update and adapt their policies, resulting in an adaptive deep MBRL method.

## 5 LoFo Replay Buffer With Contrastive State Locality

An instantiation of a LoFo replay buffer requires a definition of a local neighbourhood to an observed new sample and a way to determine which samples in the replay buffer are within that local neighbourhood. While there could be several ways to do this, in this work, we learn a state locality function using a contrastive learning technique which is then used to both define the local neighbourhood and also identify samples within that local neighbourhood.

Using contrastive learning (Hadsell et al., 2006; Dosovitskiy et al., 2014; Wu et al., 2018) we learn neural embedding representation of states such that states that are temporally closer, i.e., reachable

with fewer actions are also closer in their neural embedding representation. Specifically, we learn an embedding function $v = f_{\boldsymbol{\theta}}(s)$, where $f_{\boldsymbol{\theta}}$ is a deep neural network that maps state $s$ to an embedding vector $v$. This embedding function induces a distance metric in the state-space, such as $d_{\boldsymbol{\theta}}(s_i, s_j) = ||f_{\boldsymbol{\theta}}(s_i) - f_{\boldsymbol{\theta}}(s_j)||_2$ for states $s_i$ and $s_j$. We train the embedding function such that states that are temporally closer, have a smaller distance between the embeddings learnt and states that are temporally farther, have a relatively larger distance between them in the embedding space.

Let $s'$ represent a state that is one action away from state $s$, and $\bar{s}$ represent a set of states that are not. Let $\mathbb{D} = \{s, s', \bar{s}\}$ be a dataset of their collection. We train the embedding function to minimize the following loss function:

$$L(\mathbb{D}) = \sum_{(s, s', \bar{s}) \in D} ||f_{\boldsymbol{\theta}}(s) - f_{\boldsymbol{\theta}}(s')||_2^2 + \left(\beta - \Sigma_{\bar{s} \in \bar{s}} ||f_{\boldsymbol{\theta}}(s) - f_{\boldsymbol{\theta}}(\bar{s})||_2^2\right)^2,$$

where $\beta > 0$ is a hyperparameter. This loss function trains the embedding of states $s$ and $s'$ that are temporally next to each other to be closer by minimising their distance towards zero, while pushing the embedding of states $\bar{s}$ that are not temporally next to $s$ to be on average farther, with a cumulative squared distance value close to $\beta$, a positive number.

In our experiments, we collect trajectories using a random behavior policy at the beginning of training and use samples from that trajectory to form the dataset $\mathbb{D} = \{s, s', \bar{s}\}$ to train our embedding function $f_{\boldsymbol{\theta}}$. The learnt embedding function is then fixed and the distance between the state embeddings is used as a proxy for state locality. In complex environments with exploration challenges, a random behavior policy might be insufficient to cover the relevant state-space needed to learn a good state locality estimate. It is an important future work to figure out good ways to learn state locality in such settings.

The samples stored in the replay buffer are generally of the form $(s, a, r, s')$, representing a transition an agent experienced by taking an action $a$, from the state $s$ and moving to a state $s'$ and obtaining a reward $r$. In a LOFO replay buffer, upon taking an action in the environment, the generated experience sample is used to first gather all the samples in the local neighbourhood of the new sample. This is done by estimating the state locality using the distance $d_{\boldsymbol{\theta}}(s, s_i) = ||f_{\boldsymbol{\theta}}(s) - f_{\boldsymbol{\theta}}(s_i)||_2$ between state $s$ in the new sample and states $\{s_i\}$ corresponding to starting states in all of the samples in the replay buffer. The samples in the replay buffer whose starting state's distance to $s$, $d_{\boldsymbol{\theta}}(s, s_i) < D_{local}$ are selected. $D_{local}$ is a scalar hyperparameter that determines the size of the local neighbourhood around any given state.

If the number of samples within the local neighbourhood is equal to or above a threshold $N_{local}$, then the oldest sample in that local neighbourhood is removed. $N_{local}$ is a positive integer hyperparameter that determines the maximum number of samples that are stored in the replay buffer within any local neighbourhood of a given sample. The new sample is now added to the LOFO replay buffer.

## 6 ADAPTIVE DEEP DYNA-Q

Wan et al. (2022) successfully built an adaptive linear Dyna-Q MBRL method. However, they showed that a deep-learning-based version of the same approach failed to achieve adaptivity. They further identified inferior learned models resulting from catastrophic forgetting or interference from stale data as the key reason for the failure. In this section we show that replacing the traditional FIFO replay buffer used in Wan et al. (2022) with the LOFO replay buffer makes deep Dyna-Q adaptive.

We evaluate deep Dyna-Q with LOFO replay buffer on the LoCA setup of two domains. First, the LoCA setup of the MountainCar domain (MountainCarLoCA) used in Wan et al. (2022). Second, the LoCA setup of a variant of the simple Mini-grid domain (MiniGridLoCA). Our empirical results show that deep Dyna-Q with a LOFO replay buffer is successfully able to adapt to the local environment changes in the LoCA setup in both the domains.

MountainCarLoCA setup (Figure 2a) consists of a variation of the classical MountainCar domain, with an under-powered cart having to move up a hill. There are two terminal states, T1 at the top of the hill and T2 corresponding to the cart being at the bottom of the hill with velocity close to zero. Figure 3a shows the learning curves of different versions of deep Dyna-Q on MountainCarLoCA.

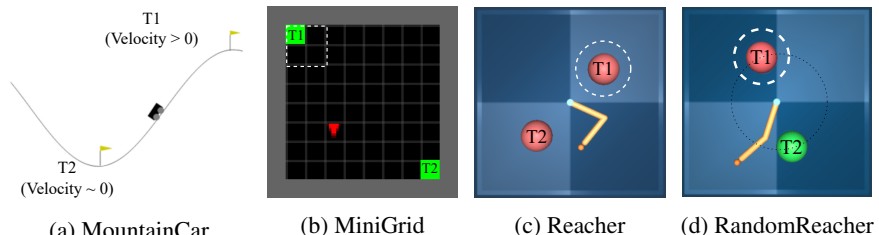



(a) MountainCar      (b) MiniGrid      (c) Reacher      (d) RandomReacher



Figure 2: Illustration of environments corresponding to the different domains used in our experiments. The dashed white lines (not visible to the agent) show the boundary of T1-zone.

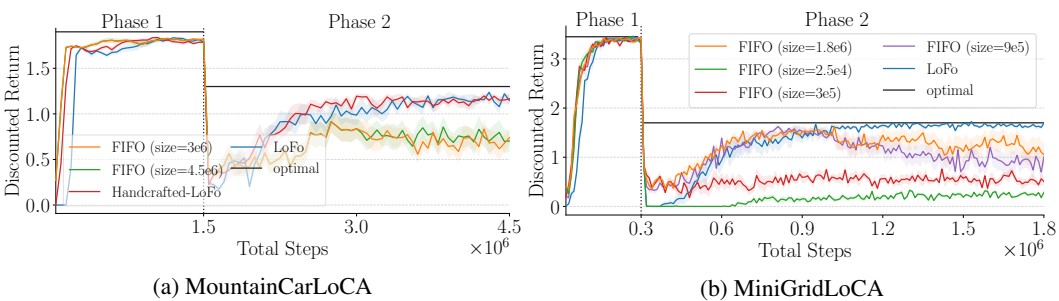



(a) MountainCarLoCA          (b) MiniGridLoCA



Figure 3: Plots showing the learning curves of deep Dyna-Q with a LoFo replay buffer (adaptive) and FIFO replay buffers of different sizes (not adaptive) on (a) MountainCarLoCA and (b) Mini-GridLoCA. Each learning curve is an average discounted return over ten runs, and the shaded area represents the standard error. The maximum possible return in each Phase is represented by a solid black line. (a) Note that the Handcrafted-LoFo refers to a variant of the LoFo replay buffer that, instead of learning the state locality function, uses a handcrafted locality function (See Appendix A.4).

The best replay buffer size for the baseline deep Dyna-Q with FIFO replay buffer is $4.5e6$, the one that stores all the samples seen so far. We observe that while all the methods reach close to optimal performance in Phase 1, only the method with LoFo replay buffer is able to adapt to reward change that happens in Phase 2 and reach close to optimal performance in Phase 2, making it an adaptive MBRL method as per the LoCA evaluation.

The two-dimensional state-space of the MountainCar domain allows visualization of properties associated with states throughout the state-space on a 2D plot. Figure 4 shows the 2D histogram of the states across the state-space whose transition samples are stored in the LoFo replay buffer and the traditional FIFO replay buffer at the end of Phase 1 and Phase 2. We observe that in a FIFO replay buffer, at the end of Phase 2, almost all of the samples in the buffer are just from a small region in the state-space, that corresponds to the T1-zone. Samples of states from other parts of the state-space that were present in Phase 1 have mostly been removed from the buffer. This leads to catastrophic forgetting of model prediction for states in parts of the state-space outside the T1-zone. In LoFo replay buffer however, even at the end of Phase 2, samples are maintained from across the entire relevant state-space, enabling maintaining accurate world model for the entire relevant state-space. Also, LoFo replay buffer stores only about $3e4$ samples in total at the end of Phase 2, compared to $4.5e6$ samples stored in the FIFO replay buffer.

The MiniGridLoCA setup has an image based high-dimensional input. The environment (Figure 2b) is an $8 \times 8$ RGB grid world with two green colored terminal states, T1 at the top left corner with a $2 \times 2$ T1-zone and T2 at the bottom-right corner. The agent is a red triangle which can choose to go straight, turn left or turn right. We use a deep convolution neural network to encode the image observations.

Figure 3b shows the learning curves of different versions of deep Dyna-Q on MiniGridLoCA. We compare deep Dyna-Q using LoFo replay buffer with the variants that use FIFO replay buffers with different buffer sizes, ranging from buffers that store all the samples seen (size = $1.8e6$) to the buffer that stores only as many samples as LoFo replay buffer uses (size = $2.56e4$). We observe that only

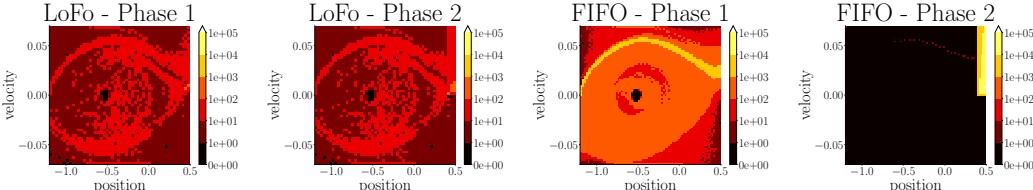

Figure 4: Histogram of the states across the state-space of MountainCarLoCA environment whose transition samples are stored in the LOFO replay buffer and the traditional FIFO replay buffer (size = $3e6$) at the end of Phase 1 and Phase 2. While LOFO replay buffer maintains samples from states throughout the relevant state-space in both Phase 1 and Phase 2, FIFO replay buffer removes almost all samples from states other than the T1-zone by the end of Phase 2.

deep Dyna-Q with LOFO replay buffer is able to adapt to changes in Phase 2 and converge to close to optimal performance. The method achieves adaptivity while storing only around $2.56e4$ samples in the replay buffer, two orders of magnitude less samples compared to the best performing FIFO replay buffer based method which is of size $1.8e6$. Note that among the methods that use FIFO replay buffers, two methods with largest buffer sizes, one with a buffer of size $9e5$ and the other that stores all the samples with a size of $1.8e6$ are initially able to reach close to optimal performance in Phase 2, but their performance degrades over time afterwards. This is because, the total number of distinct states in the MiniGridLoCA environment are less (256 to be precise) and therefore samples from all over the state-space from Phase 1 stay longer in a large replay buffer. There is a sweet spot when there are still samples from throughout the state-space from Phase 1, while the proportion of samples from Phase 2 that capture the reward change in the T1-zone are much higher than the stale rewards of T1 from Phase 1 avoiding any serious interference. This can lead to a performance that is temporarily close to optimal performance but quickly degrades as samples from Phase 1 are removed in the FIFO buffer and catastrophic forgetting happens. Additional details on all the deep Dyna-Q experiments are provided in Appendix A and B.

## 7    ADAPTIVE PLANET AND DREAMERV2

To see if more complex methods can be made adaptive as well by using a LoFo replay buffer, we applied it to the deep MBRL methods PlaNet (Hafner et al., 2019b) and DreamerV2 (Hafner et al., 2020) and evaluated their performance. PlaNet and DreamerV2 use a recurrent model for reward and transition predictions which require sample-sequences to make updates to rather than individual samples. Therefore, we first need to extend the concept of the LOFO replay buffer to sample-sequences. On a high level, what we aim to achieve is that updates made to the world model come from states spread out more-or-less equally across the state-space. In the case of sample-sequences, we try to approximate this by ensuring that the start-state of each sequence is drawn approximately at random across the state-space.

Our approach to achieve good coverage of sequence start-states is to use two separate buffers: a *state-buffer* which is curated similarly to the LOFO replay buffer, and a *trajectory-buffer* storing observed trajectories. For now, assume that the trajectory-buffer stores all observed trajectories. In the Appendix E, we show how the trajectory-buffer can be bounded to a maximum sample-size of $B \cdot N$, where $B$ is the size of the state-buffer and $N$ is the size of a sample-sequence used for updates. When a new sample $(s_t, a_t, r_t, s_{t+1})$ is observed, it is appended to the trajectory-buffer, while state $s_t$ is added to the state-buffer. Crucially, if the state-buffer contains more states from the local neighborhood of state $s_t$ than some threshold amount, the oldest state from this neighborhood is removed from the state-buffer. Each state in the state-buffer points to a copy of itself in the trajectory-buffer. When a state $s_i$ is removed from the state-buffer, the corresponding reward $r_i$ in the trajectory buffer is replaced by $None$, indicating that these rewards should no longer be used for training reward-predictions. The world model is updated by first sampling a random state $s_u$ from the state-buffer and then drawing a sample sequence of size $N$ from the trajectory-buffer, starting with state $s_u$. Whenever the reward is $None$, the loss of the corresponding reward prediction is set to 0.

We evaluate our modified versions of PlaNet and DreamerV2 on the LoCA setup applied to two domains. First, a variation on the Reacher domain, called ReacherLoCA, introduced by Wan et al. (2022). And second, a more complex extension of ReacherLoCA which we refer to as the RandomizedReacherLoCA. For both domains, we observe that LOFO replay buffer results in adapting to the local reward change in T1 and learning a sufficiently accurate reward model throughout the relevant state-space in Phase 2, reaching close to optimal performance.

The ReacherLoCA setup consists of a variation of the Reacher domain Tassa et al. (2018). It is a continuous-action domain with 64 x 64 RGB images as observations. The environment has two targets corresponding to T1 and T2 fixed at the top right and the bottom left quadrants respectively (Figure 2c). The agent controls the angular velocity of two connected bars to reach a target and remain at the target till the episode ends (1000 time steps). The RandomizedReacherLoCA (Figure 2d) setup is an extension of the ReacherLoCA where the location of targets in the Reacher environment can vary each episode along a circle around the center, while still being opposite to each other. The target T2 is colored differently from T1 for the agent to be able to differentiate between them.

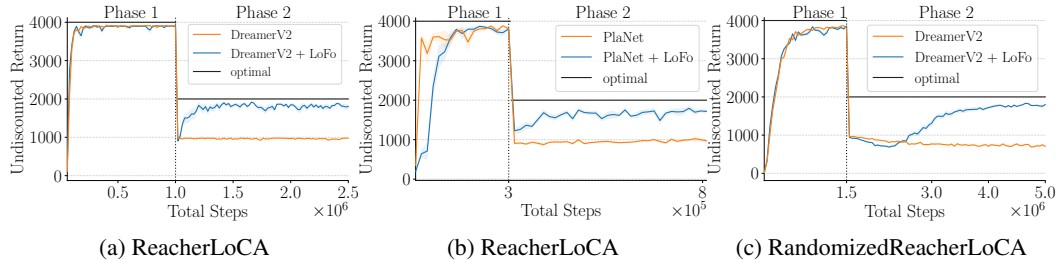

Figure 5: Plots showing the learning curves of DreamerV2 and PlaNet with a LOFO replay buffer (adaptive) and with FIFO replay buffers (not adaptive) on a, b) ReacherLoCA, and c) RandomizedReacherLoCA. Each learning curve is an average undiscounted return over ten runs, and the shaded area represents the standard error. The maximum possible return in each phase is represented by a solid black line.

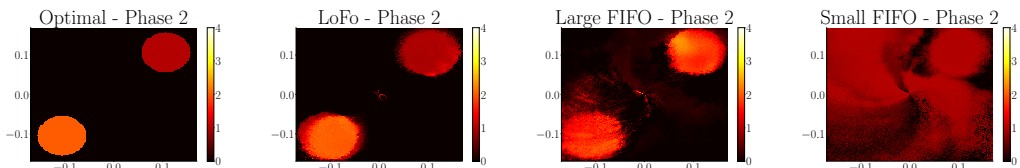

Figure 6: Visualization of the estimated rewards from the DreamerV2 agent's reward model at the end of Phase 2. Each point on the heatmap represents the agent's position in the Reacher environment.

Figure 5 shows the learning curves of DreamerV2 and PlaNet on ReacherLoCA, and DreamerV2 on RandomizedReacherLoCA. We observe that in both the setups all the methods reach close to optimal performance in Phase 1. However, in Phase 2 only DreamerV2 and PlaNet with a LOFO replay buffer is able to adapt to the environment change, making them adaptive deep MBRL methods. DreamerV2 and PlaNet with the traditional FIFO replay buffer fail to adapt in Phase 2.

In Figure 6 we visualize the reward predictions of the different DreamerV2 methods. We observe that DreamerV2 that uses the LOFO replay buffer for learning its reward model has adapted its reward predictions for target T1 (top right) correctly to around +1 in Phase 2. When we use a large FIFO replay buffer, we observe that the reward for target T1 at the end of Phase 2 is overestimated to around +2.5 because of the interference of stale samples from Phase 1. On the other hand when we use a small FIFO replay buffer, DreamerV2's reward prediction at the end of Phase 2 for T1 is accurate around +1, but the model has completely forgotten the reward for target T2, and other parts of the state-space outside the T1-zone. Additional details on all the PlaNet and DreamerV2 experiments are provided in Appendix C and D.

## 8 RELATED WORK

Besides the relation to the work of Wan et al. (2022), discussed in Section 6, this work has connections with several other lines of work. The method used for learning the state locality function (Section 5) in this work is inspired by the work of Hartikainen et al. (2019). They propose a method to learn a policy-specific distance-function that is a measure of the expected number of time steps to reach the goal state from a given state. And they use this to shape the reward function.

Several works have proposed modifications to the traditional FIFO replay buffer to handle various challenges. The work that is most related to ours is that of Purushwalkam et al. (2022). They focused on the setting of self-supervised learning from a continual stream of unsupervised data. And to address catastrophic forgetting they proposed to use a replay buffer that removes the most correlated samples when the buffer is full. For the purpose of adaptation this is not a good strategy, however, as incorrect, out-of-date data can remain in the buffer for a long time.

LOFO replay buffer provides a mechanism for adding and removing samples from a replay buffer. This determines the distribution of samples that are stored in the replay buffer. Various works in the past have developed different strategies for sampling or weighting samples in the replay buffer, in order to make the training more efficient. These include prioritizing samples that 1) have high temporal difference (TD) error (Schaul et al., 2015), 2) are closest to the current state the agent is in (Sun et al., 2020), and 3) have low TD target uncertainty (Kumar et al., 2020; Lee et al., 2021). These methods are complementary to our work and can be implemented on a LOFO replay buffer.

## 9 LIMITATIONS AND FUTURE WORK

The general strategy behind the replay buffer variation we presented boils down to forgetting samples that are *spatially close, but temporally far* from currently observed samples. We believe this to be a good general strategy to achieve effective adaptation in a non-stationary world. However, the specific implementation of this principle will differ depending on the problem type and the non-stationarity considered. In this paper, we considered only reward non-stationarity and domains where exploration is easy. In this scenario, learning a locality function during the initial learning phase and keeping it fixed thereafter is sufficient. However, when the transition dynamics is non-stationary as well, the locality function needs to be maintained and updated across time as well, as the distances between individual states can change over time.

Now that we have shown that adaptive deep MBRL is possible in principle, a logical next step for future work is scaling up these methods to larger and harder domains. Because, while DreamerV2 has shown to be able to achieve good single-task performance on such domains, it is not a given that our replay buffer variation is sufficient for making it adaptive for such domains as well. For example, the ReacherLoCA we considered has a fairly small decision horizon (i.e., how far an agent needs to plan ahead to construct a good policy—for ReacherLoCA it takes on average about 25 actions to reach a goal state). Achieving adaptivity for longer decision horizons puts higher demands on the planning routine as well (see Wan et al. (2022) for planning-related pitfalls that impede adaptivity), so it might be needed to make changes to DreamerV2's planning routine as well.

## 10 CONCLUSION

In this work, we considered one of the key features of model-based behavior: the ability to adapt to local environmental changes. In previous work, it has shown that tabular and linear model-based methods are able to achieve this form of adaptivity. However, deep model-based methods struggle due to an interplay between catastrophic forgetting and interference. To address the challenges with deep model-based methods, we proposed the LOFO replay buffer, whose samples are approximately spread equally across the state-space. Furthermore, we conducted various experiments to show that utilizing the LOFO replay buffer with deep MBRL methods can make them adapt effectively to local changes in the reward function. This is the first time–to the best of our knowledge–that this type of adaptivity has been shown for deep MBRL methods. This is an important step towards more practical real-world application of RL, since the stationary assumption does not always apply there.

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

# A   Additional Details for the Experiments on MountainCarLoCA Setup

## A.1   Experiment Setup

| | | |
|---|---|---|
| Initial distributions | Phase 1 training | Uniform distribution over the entire state-space |
| | Phase 1 evaluation | Uniform distribution over a small region |
| | Phase 2 training | Uniform distribution over states within T1-zone |
| | Phase 2 evaluation | Uniform distribution over a small region |
| Training steps | Phase 1 steps | $1.5 \times 10^6$ |
| | Phase 2 steps | $3 \times 10^6$ |
| Other details | Maximum number of steps before an episode terminates | 500 |
| | Training steps between two evaluations | $10^4$ |
| | Number of runs | 10 |
| | Number of evaluation episodes | 10 |

Table 1: Experiment setup for testing the Deep Dyna-Q on the MountainCarLoCA domain.

The MountainCarLoCA was first introduced by Van Seijen et al. (2020), which is a variant of the well-known MountainCar environment. T1 is located at the top of the mountain (position $> 0.5$, and velocity $> 0$), and T2 is located at the valley ($(\text{position} + 0.52)^2 + 100 \times \text{velocity}^2 \leq 0.07^2$). The T1-zone contains all the states within $0.4 \leq \text{position} \leq 0.5$ and $0 \leq \text{velocity} \leq 0.07$. The discount factor for this environment is $\lambda = 0.99$. And lastly, for each evaluation, the agent is initialized roughly at the middle of T1 and T2 ($-0.2 \leq \text{position} \leq -0.1$ and $-0.01 \leq \text{velocity} \leq 0.01$). Table 1 shows the experiment setup we used to evaluate the deep Dyna-Q agent's adaptivity under the LoCA setup.

## A.2   Hyperparameters

Table 2 shows the final values of the hyperparameters that were used in the LoFo replay buffer for generating the results presented in Figure 3a. We searched over $\beta \in \{1, 10\}$, $D_{local} \in \{0.01, 0.005\}$, and $N_{local} \in \{1, 2, 5, 10\}$.

We used the same version of the deep Dyna-Q algorithm used in Wan et al. (2022). However, instead of having separate neural networks for different actions in each part of the model (dynamics, reward, and termination models), we use just one network and concatenate the action with the output of the middle layer (the one that has 63 output units) and then feed it to the next layer. Table 3 summarizes the important hyperparameters for the deep Dyna-Q method on the MountainCarLoCA. Both the deep Dyna-Q agent that used the LoFo replay buffer and the baseline agents in Figure 3a used these hyperparameters, and they only differ in the choice of the replay buffer. Wan et al. (2022) found that using the same replay buffer size for learning the model and planning worked the best. Therefore, we followed the same.

## A.3   Additional Results for The Deep Dyna-Q

We compare the reward models of the Dyna-Q agents that use the LoFo replay buffer and the traditional FIFO replay buffer by visualizing their prediction of rewards over all states in the MountainCarLoCA domain at the end of each phase in Figure 7. We observe that the reward model of the agent that uses the LoFo replay buffer can estimate the rewards correctly in both phases. However, this is not true for the agent that uses the traditional FIFO replay buffer since it fails to predict the correct rewards associated with T2 due to catastrophic forgetting.

| Embedding network architecture | MLP:$[64 \times 64 \times 64 \times 16]$, Activation Function: *tanh* |
|---|---|
| Optimizer | Adam, learning rate: $10^{-4}$ |
| $\beta$ | 10 |
| Number of negative samples | 128 |
| Mini-batch size | 32 |
| Total number of random steps for creating dataset $\mathbb{D}$ | 100000 |
| Number of training epochs | 5 |
| $D_{local}$ | 0.005 |
| $N_{local}$ | 1 |

Table 2: Hyperparameters used for the LOFO replay buffer on the MountainCarLoCA domain.

| Neural networks | Dynamics model | MLP with *tanh*, $[64 \times 64 \times 63 \times 64 \times 64]$, |
|---|---|---|
| | Reward model | MLP with *tanh*, $[64 \times 64 \times 63 \times 64 \times 64]$, |
| | Termination model | MLP with *tanh*, $[64 \times 64 \times 63 \times 64 \times 64]$, |
| | Action-value estimator | MLP with *tanh*, $[64 \times 64 \times 64 \times 64]$, |
| Optimizer | Value optimizer | Adam, learning rate: $5 \times 10^{-6}$ |
| | Model optimizer | Adam, learning rate: $5 \times 10^{-5}$ |
| Other details | Exploration parameter | Epsilon greedy $\epsilon = 0.5$ |
| | Number of random steps before training | 50000 |
| | Target network update frequency | 500 |
| | Number of model learning steps | 5 |
| | Number of planning steps | 5 |
| | Mini-batch size of model learning | 32 |
| | Mini-batch size of planning | 32 |

Table 3: Hyperparameters used for the deep Dyna-Q on the MountainCarLoCA domain.

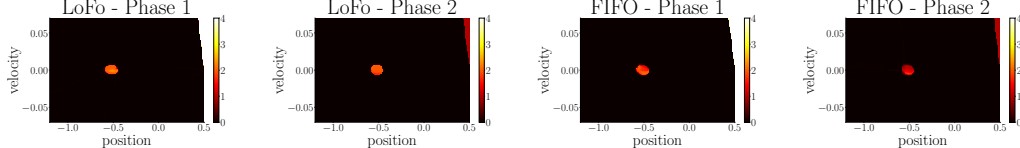

Figure 7: Visualization of the estimated rewards from the deep Dyna-Q agent's reward model at the end of each phase. In each heatmap, the $x$ axis represents the agent's position, and the $y$ axis represents its velocity.

A.4 HANDCRAFTED STATE LOCALITY FUNCTION

Additionally, we tested another variant of the LOFO replay buffer that uses a handcrafted state locality function instead of using the learnt contrastive state locality function. Since the Mountain-CarLoCA domain's state-space is 2-dimensional, where the first dimension indicates the position and the second one the velocity of the car ($s : (x, v)$), we can scale the range of the velocities to match the scale of the positions and use the euclidean distance as the locality function:

$$d_{\text{handcrafted}}(s_i, s_j) = \sqrt{(x_i - x_j)^2 + 150 \times (v_i - v_j)^2}$$

The learning curves of the deep Dyna-Q agent that uses the LOFO replay buffer with the handcrafted state locality function in the MountainCarLoCA domain is presented in Figure 3a. We observe that the LOFO replay buffer with the learnt state locality function is able to match the performance of the buffer with the handcrafted state locality function.

# B  ADDITIONAL DETAILS FOR THE EXPERIMENTS ON THE MINIGRIDLOCA SETUP

## B.1  EXPERIMENT SETUP

| | | |
|---|---|---|
| Initial distributions | Phase 1 training | Uniform distribution over the entire state-space |
| | Phase 1 evaluation | Uniform distribution over the entire state-space |
| | Phase 2 training | Uniform distribution over states within T1-zone ($2 \times 2$ subgrid) |
| | Phase 2 evaluation | Uniform distribution over the entire state-space |
| Training steps | Phase 1 steps | $3 \times 10^5$ |
| | Phase 2 steps | $1.5 \times 10^6$ |
| Other details | Maximum number of steps before an episode terminates | 100 |
| | Training steps between two evaluations | $10^4$ |
| | Number of runs | 10 |
| | Number of evaluation episodes | 10 |

Table 4: Experiment setup for testing the Deep Dyna-Q method on the MiniGridLoCA domain.

The implementation of the MiniGridLoCA domain is done using the MiniGrid python package (Chevalier-Boisvert et al., 2018) . Table 4 shows the experiment setup we used to evaluate the deep Dyna-Q agent's adaptivity under the LoCA setup.

## B.2  HYPERPARAMETERS

Table 5 shows the final hyperparameters setting used in the LOFO replay buffer for generating the results presented in Figure 3b. Furthermore, we only searched over $N_{local} \in \{50, 75, 100, 150\}$. While we started with the same $\beta$ and $D_{local}$ as the MountainCarLoCA, we found that by decreasing the $D_{local}$ to 0.001, the learned state locality function can make a clear distinction between all possible states (256).

The algorithmic design of the deep Dyna-Q agent we used for the MiniGridLoCA domain is the same as that used for the MountainCarLoCA. We only changed the neural network architecture for the model and the action-value estimator. Table 6 summarizes the architecture of the neural networks. Note that we first encode a given state to a low-dimensional vector for the various parts of the model (dynamics, reward, and termination model). Then, we concatenate the given action to the resulting vector and feed it to the MLP layers. This is mainly due to the fact that we did not want to

| | |
|---|---|
| Embedding network architecture | CNN: (Channels:$[32 \times 64 \times 64]$ Kernel Sizes:$[8 \times 3 \times 3]$ Strides:$[4 \times 2 \times 2]$), Followed by MLP:$[512 \times 16]$, Activation Function: *relu* |
| Optimizer | Adam, learning rate: $10^{-4}$ |
| $\beta$ | 10 |
| Number of negative samples | 128 |
| Mini-batch size | 32 |
| Total number of random steps for creating dataset $\mathbb{D}$ | 25000 |
| Number of training epochs | 5 |
| $D_{local}$ | 0.001 |
| $N_{local}$ | 1 |

Table 5: Hyperparameters used for the LoFo replay buffer on the MiniGridLoCA domain.

| | | |
|---|---|---|
| Neural networks | Dynamics model | CNN: (Channels:$[32 \times 64 \times 64]$ Kernel Sizes:$[8 \times 3 \times 3]$ Strides:$[4 \times 2 \times 2]$), Followed by Transposed CNN: (Channels:$[64 \times 32 \times 3]$ Kernel Sizes:$[6 \times 6 \times 5]$ Strides:$[1 \times 4 \times 3]$), Activation Function: *relu* |
| | Reward model | CNN: (Channels:$[32 \times 64 \times 64]$ Kernel Sizes:$[8 \times 3 \times 3]$ Strides:$[4 \times 2 \times 2]$), Followed by MLP:$[512]$, Activation Function: *relu* |
| | Termination model | CNN: (Channels:$[32 \times 64 \times 64]$ Kernel Sizes:$[8 \times 3 \times 3]$ Strides:$[4 \times 2 \times 2]$), Followed by MLP:$[512]$, Activation Function: *relu* |
| | Action-value estimator | CNN: (Channels:$[32 \times 64 \times 64]$ Kernel Sizes:$[8 \times 3 \times 3]$ Strides:$[4 \times 2 \times 2]$), Followed by MLP:$[512]$, Activation Function: *relu* |

Table 6: Neural networks' architecture for the deep Dyna-Q on the MiniGridLoCA domain.

keep separate networks for each action as it was done in Wan et al. (2022). Finally, Table 7 shows the final hyperparameters used to generate the learning curves in Figure 3b, when the agent used the LoFo replay buffer and the traditional FIFO replay buffer.

### B.3 ADDITIONAL RESULTS FOR THE DEEP DYNA-Q

Figure 8 shows the learning curves of our search over different values of $N_{local}$ for the deep Dyna-Q agent using the LoFo replay buffer. Since the locality function makes a clear distinction between the states of the MiniGridLoCA, using a specific $N_{local}$ means that the LoFo replay buffer stores

| Optimizer | Value optimizer | Adam, learning rate: $6.25 \times 10^{-5}$ |
|---|---|---|
| | Model optimizer | Adam, learning rate: $10^{-4}$ |
| Other details | Exploration parameter | Epsilon greedy $\epsilon = 0.5$ |
| | Number of random steps before training | 2000 |
| | Target network update frequency | 5000 |
| | Number of model learning steps | 1 |
| | Number of planning steps | 1 |
| | Mini-batch size of model learning | 128 |
| | Mini-batch size of planning | 128 |

Table 7: Final hyperparameters for the deep Dyna-Q on the MiniGridLoCA domain.

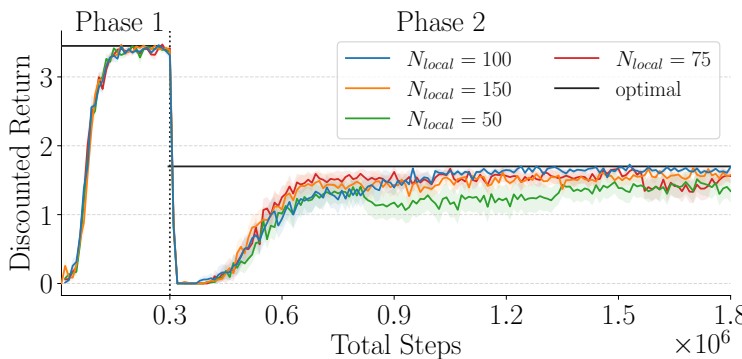

Figure 8: Learning curves for the Dyna-Q agent that uses the LOFO replay buffer with different $N_{local}$ values ($D_{local} = 0.001$), each averaged over 10 random seeds.

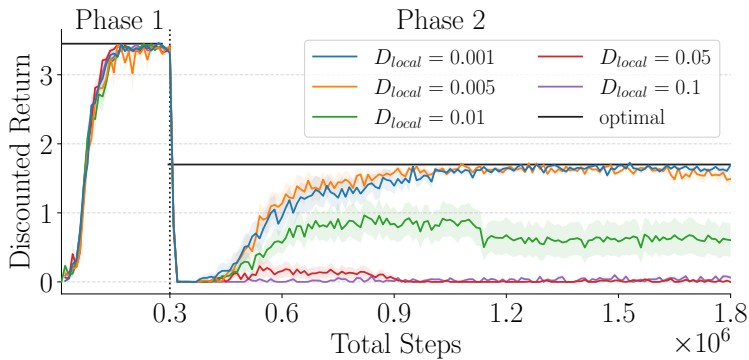

Figure 9: Learning curves for the Dyna-Q agent that uses the LOFO replay buffer with different $D_{local}$ values ( $N_{local} = 100$), each averaged over 10 random seeds.

exactly $N_{local}$ samples per each state. Hence, an interesting observation from Figure 8 is that higher $N_{local}$ results in higher performance in Phase 2.

Figure 9 shows the learning curves of our search over different values of $D_{local}$ for the deep Dyna-Q agent using the LOFO replay buffer when $N_{local} = 100$. We observe that small values $D_{local}$ result in successful adaptation in Phase 2.

Additionally, we visualized the estimated rewards from the deep Dyna-Q agent's reward model when using the LOFO replay buffer and the traditional FIFO replay buffer at the end of each phase in Figure 10. To create each heatmap, we first generated the estimated rewards by placing the agent in a given state (a specific cell and direction) and taking the going straight action. This gave us four

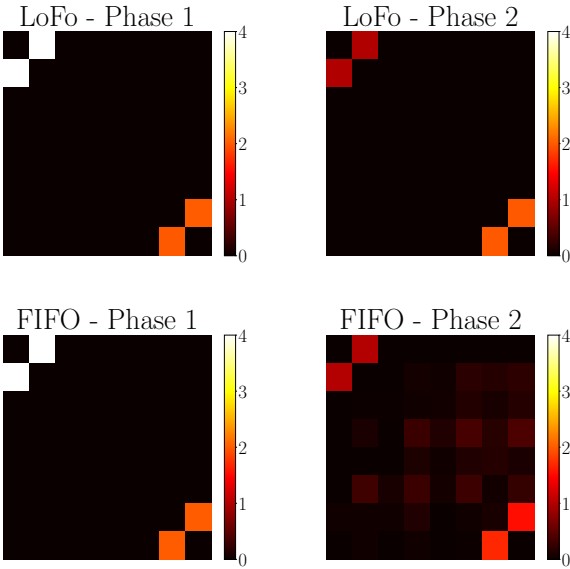

Figure 10: Visualization of the estimated rewards from the deep Dyna-Q agent's reward model at the end of each phase. Each point on the heatmap represents the agent's position in $8 \times 8$ grid of the MiniGridLoCA environment.

different 2D heatmaps, each representing a particular direction. Then, we summed the estimated rewards over the four possible directions and created the final heatmap.

From Figure 10, we observe the deep Dyna-Q agent that uses the LOFO replay buffer can predict the correct summation of the rewards during each phase. In contrast, the agent that uses the traditional FIFO replay buffer can only make accurate predictions in Phase 1. The inability of this agent to correctly predict the summation of the rewards is due to forgetting. Note that we used the agent with the traditional FIFO replay buffer of size $3e5$ in Figure 10.

## C    ADDITIONAL DETAILS FOR THE EXPERIMENTS ON THE REACHERLOCA SETUP

### C.1    EXPERIMENT SETUP

| | | |
|---|---|---|
| Initial distributions | Phase 1 training | Uniform distribution over the entire state-space |
| | Phase 1 evaluation | Uniform distribution over the entire states outside T1-zone |
| | Phase 2 training | Uniform distribution over states within T1-zone |
| | Phase 2 evaluation | Uniform distribution over the entire states outside T1-zone |
| Training steps | Phase 1 steps | $3 \times 10^5$ |
| | Phase 2 steps | $5 \times 10^5$ |
| Other details | Number of steps before an episode terminates | 1000 |
| | Training steps between two evaluations | 15000 |
| | Number of runs | 10 |
| | Number of evaluation episodes | 5 |

Table 8: Experiment setup for testing the PlaNet method on the ReacherLoCA domain.

| Initial distributions | Phase 1 training | Uniform distribution over the entire state-space |
|---|---|---|
| | Phase 1 evaluation | Uniform distribution over the entire states outside T1-zone |
| | Phase 2 training | Uniform distribution over states within T1-zone |
| | Phase 2 evaluation | Uniform distribution over the entire states outside T1-zone |
| Training steps | Phase 1 steps | $10^6$ |
| | Phase 2 steps | $1.5 \times 10^6$ |
| Other details | Number of steps before an episode terminates | 1000 |
| | Training steps between two evaluations | 10000 |
| | Number of runs | 10 |
| | Number of evaluation episodes | 8 |

Table 9: Experiment setup for testing the DreamerV2 method on the ReacherLoCA domain.

Tables 8 and 9 show the experiment setup we used to evaluate the PlaNet and the DreamerV2 agents' adaptivity under the LoCA setup respectively. It is worth mentioning that Wan et al. (2022) stayed as close as possible to the original Reacher domain (Tassa et al., 2018) when creating the ReacherLoCA to facilitate reusing the best hyperparameters previously used as much as possible. A transition to the targets in the ReacherLoCA domain does not terminate the episode and the agent keeps receiving rewards until 1000 timesteps. Note that having targets instead of terminal states does not affect the requirements of the LoCA setup.

## C.2 HYPERPARAMETERS

| Embedding network architecture | CNN: (Channels:$[32 \times 64 \times 128 \times 256]$ Kernel Sizes:$[4 \times 4 \times 4 \times 4]$ Strides:$[2 \times 2 \times 2 \times 2]$), Followed by MLP:$[512 \times 64, 32]$, Activation Function: *relu* |
|---|---|
| Optimizer | Adam, learning rate: $10^{-4}$ |
| $\beta$ | 50 |
| Number of negative samples | 128 |
| Mini-batch size | 32 |
| Total number of random steps for creating dataset $\mathbb{D}$ | $10^5$ |
| Number of training epochs | 5 |
| $D_{local}$ | 0.05 |
| $N_{local}$ | 10 |

Table 10: Hyperparameters used for the LOFO replay buffer on the ReacherLoCA domains.

Table 10 shows the values of the hyperparameters used in the LOFO replay buffer for generating the results presented in Figures 5a and 5b. Furthermore, we searched over different values of $N_{local} \in \{2, 5, 10, 20, 40, 80\}$. For both PlaNet and DreamerV2 agents the best setting in our experiments is $N_{local} = 10$.

For the rest of the hyperparameters, we use the hyperparameter values used by Wan et al. (2022) for the DreamerV2 agent for both when it uses the LOFO replay buffer and the traditional replay buffer. Furthermore, instead of having an entropy regularizer for the exploration, we add noise to actions (the default value used in the original Dreamer method (Hafner et al., 2019a)) since the ReacherLoCA domain is a relatively simple environment.

For the PlaNet agent, we perform a hyperparameter search only for the learning rate ($lr \in \{3 * 10^{-4}, 10^{-4}\}$). The best learning rate for the agent that uses the LoFo replay buffer is $3 * 10^{-4}$. Other hyperparameters are the same as the best setting in Wan et al. (2022).

## C.3  ADDITIONAL RESULTS FOR PLANET

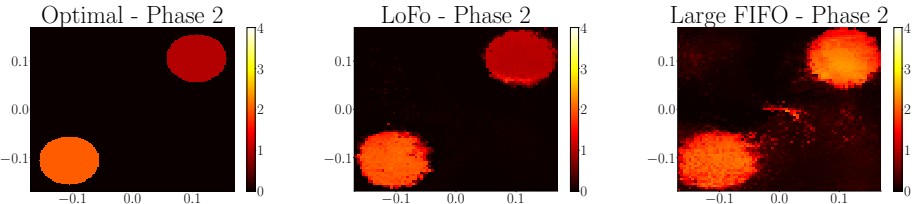

Figure 11: Visualization of the estimated rewards from the PlaNet agent's reward model at the end of Phase 2. Each heatmap's $x$ and $y$ axes represent the agent's position in the ReacherLoCA domain.

In Figure 11, we visualize the reward predictions of the PlaNet agents that use the LoFo replay buffer and the traditional FIFO replay buffer. We also added the true reward visualization (as optimal) for reference. The results are similar to those of DreamerV2's (Figure 6).

## D  ADDITIONAL DETAILS FOR THE EXPERIMENTS ON THE RANDOMIZEDREACHERLOCA SETUP

## D.1  EXPERIMENT SETUP AND HYPERPARAMETERS

| | | |
|---|---|---|
| Initial distributions | Phase 1 training | Uniform distribution over the entire state-space |
| | Phase 1 evaluation | Uniform distribution over the entire states outside T1-zone |
| | Phase 2 training | Uniform distribution over states within T1-zone |
| | Phase 2 evaluation | Uniform distribution over the entire states outside T1-zone |
| Training steps | Phase 1 steps | $1.5 \times 10^6$ |
| | Phase 2 steps | $3.5 \times 10^6$ |
| Other details | Number of steps before an episode terminates | 1000 |
| | Training steps between two evaluations | 10000 |
| | Number of runs | 10 |
| | Number of evaluation episodes | 8 |

Table 11: Experiment setup for testing the DreamerV2 method on the RandomizedReacherLoCA domain.

In the RandomizedReacherLoCA, the location of the red target (T1) is randomly taken from the circle centered in the center of the state-space (the dotted black circle in Figure 2d). And then, the green target (T2) is placed at the opposite end of the circle. Table 11 shows the experiment setup we used to evaluate the DreamverV2 agent's adaptivity under the LoCA setup.

The best hyperparameter setting for the LoFo replay buffer is exactly as what is mentioned in Table 10, except that $N_{local} = 2$ (we searched among $N_{local} \in \{1, 2, 5, 10\}$). Otherwise, we use the same hyperparameters for the DreamerV2 agent as used in the ReacherLoCA setup.

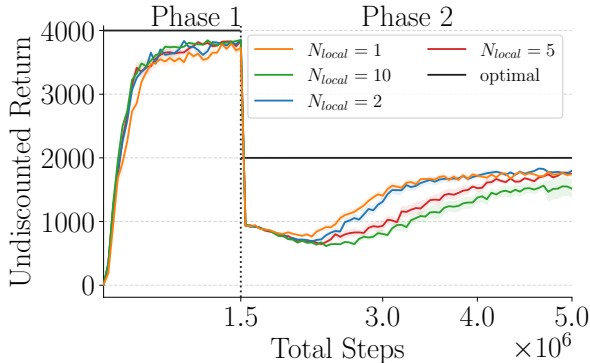

Figure 12: Learning curves for the DreamerV2 agent that uses the LOFO replay buffer with different $N_{local}$ values ($D_{local} = 0.05$), each averaged over 10 random seeds.

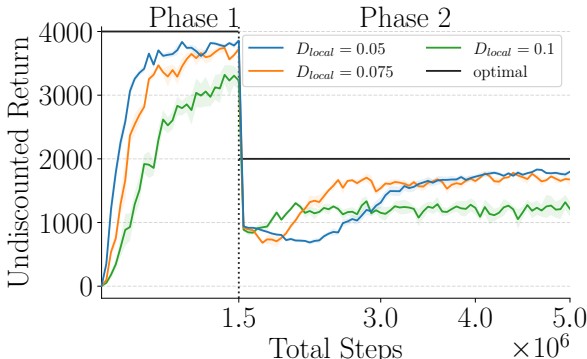

Figure 13: Learning curves for the DreamerV2 agent that uses the LOFO replay buffer with different $D_{local}$ values ($N_{local} = 2$), each averaged over 5 random seeds.

## D.2 ADDITIONAL RESULTS FOR THE DREAMER

Figure 12 and 13 shows the learning curves of our search over different values of $N_{local}$ and $D_{local}$ for the DreamerV2 agent using the LOFO replay buffer. Similar to the MiniGridLoCA domain, we observe that small values of $D_{local}$ result in successful adaptation in Phase 2 for the Randomize-dReacherLoCA.

# E    LoFo Replay Buffer with Recurrent Models

In this section, we show that *trajectory-buffer* can be bounded to a maximum sample-size of $B \times N$ in practice, where $B$ is the size of the state-buffer and $N$ is the size of a sample-sequence used for updates.

As mentioned in Section 7, upon removing $s_i$ from the state-buffer, $r_i$ is set to $None$ in the trajectory-buffer and the agent never uses a sample-sequence starting from $s_i$. Now, if we look closely at the trajectory-buffer, only sample-sequences starting with states $s_k$ where $k \in (i - N, i]$ contains the $(s_i, a_i, r_i = None, s_{i+1})$. Conceptually, we can remove $(s_i, a_i, r_i, s_{i+1})$ from the trajectory-buffer if $s_k, \forall k \in (i - N, i]$ are removed from the state-buffer, because in this case there remains no sample-sequence containing $(s_i, a_i, r_i, s_{i+1})$ for training the agent. By doing so, we argue that no $N$ consecutive $None$ rewards can be found in the trajectory buffer. Because in that case, the last $None$ reward belongs to no valid sample-sequence, and therefore, it should have been removed from the trajectory-buffer.

Given that for each state in the state-buffer, we know their corresponding reward in the trajectory-buffer is not $None$, in the worst-case scenario, there can be at most $N - 1$ $None$ rewards after such samples in the trajectory-buffer. Hence, by counting them as well, each sequence in the trajectory-buffer starting from a state in the state-buffer can be at most of the length $N$. Therefore, the total number of samples stored in the trajectory-buffer would be at most $B \times N$.

