# OpenReview forum: "Replay Buffer with Local Forgetting for Adaptive Deep Model-Based Reinforcement Learning"
_ICLR.cc/2023/Conference — Submitted to ICLR 2023_

### Official Review · Reviewer_oofa · 2022-10-24

**Confidence:** 4
**Correctness:** 4
**Technical Novelty And Significance:** 3
**Empirical Novelty And Significance:** 3
**Recommendation:** 6

**Clarity, Quality, Novelty And Reproducibility:**

The paper is clearly written and executed well. The idea is simple but interesting for the studied problem.

**Strength And Weaknesses:**

The paper is very well and clearly written. The proposed idea is rather simple - which is not a negative aspect - and makes conceptually sense to solve the studied problem. It is shown that the proposed method increases performance in the environments used for evaluation. However, as also mentioned by the authors, the environments are rather simple and it is not clear if the results would transfer to more interesting environments. Currently, the similarity function is trained on random trajectories collected at the beginning of the training. This will not work for more complicated environments, where most parts of the state space have not been observed in the random trajectories. This point limits the significance of the work. Further, while domain adaptation is an important topic it is only studied in the context of one specific benchmark that accounts for scenarios of domain adaptation but not for others.


Are the random trajectories used to train the similarity function counted as environment steps in the shown plots?

Minor things:
p.5 the 5th to last row has an unnecessary space before a comma
I find it irritating that there is textof the main manuscript between Fig.2 and Fig. 3. I suggest the Figures should be just underneath each other.

**Summary Of The Paper:**

The paper studies model-based reinforcement learning in the context of changing environments. More specifically, the algorithms are evaluated in the Local Change Adaptation (LoCA) setting. Well known methods like Dreamer struggle in this setting. The paper proposes a simple method that modifies the first-in-first-out rule of the replay buffer and instead removes old data if it is very similar to a newly added state. The similarity is measured by a function learned through contrastive learning. The proposed method shows improved results over the baseline.

**Summary Of The Review:**

The proposed method is interesting and the quality of the paper is very good. However, the limited evaluation makes the paper less significant. Still, it is a good first step and hence the paper is for me just over the acceptance threshold.

---

> ### Author Response · Authors · 2022-11-12
> **Response to Reviewer oofa**
>
> We really appreciate your valuable feedback. Furthermore, we are happy that you think our proposed method is simple and sensible for the studied problem. We provide further clarifications for your questions here:
>
> **[Limitations on the significance of the work]**
>
> *Re: “Currently, the similarity function is trained on random trajectories collected at the beginning of the training. This will not work for more complicated environments, where most parts of the state space have not been observed in the random trajectories. This point limits the significance of the work.”*
>
> The key contribution of our work is the idea of forgetting the oldest sample that is close to the current observed sample. We believe this to be a good general idea for learning world models that are approximately accurate across the relevant state-space, with local changes accounted for quickly, therefore enabling effective adaption. The specific implementation of this idea will differ depending on the problem type and the non-stationarity considered. In our work, where we considered domains with only reward changes and easy exploration, learning the locality function with random trajectories at the beginning and keeping it fixed is sufficient. However, as the reviewer rightly points out, in more complicated environments, random trajectories at the beginning will not be sufficient to learn a good locality function, and one would need better exploration strategies or even alternate methods to learn the locality function, such as incrementally over time. Also note that when the transition dynamics change as well, the locality function needs to be updated over time, as the distances between individual states can change over time. Implementing the general idea on more complicated domains is an important and logical next step. We, however, believe that does not limit the significance of our current work that introduces the general idea and has shown that deep adaptive MBRL is possible with this idea, where current popular MBRL methods fail.
>
>
> **[Experiment detail]**
>
> *Re: “Are the random trajectories used to train the similarity function counted as environment steps in the shown plots?”*
>
> For simpler illustration, we initially did not count the random steps for training the locality function (for all of our experiments), also because they are much smaller than the total number of environment steps in Phase 1. Then, however, we changed our deep Dyna-Q experiments to count them as well, to show one example. We will make this clear in the paper. Here are the details:
>
>
> | Setup   |      Steps for training the locality function      |  Total steps in Phase 1 |
> |----------|:-------------:|------:|
> | MountainCarLoCA |  $1e5$ | $1.5e6$ |
> | MiniGridLoCA |    $2.5e4$   |   $3e5$ |
> | ReacherLoCA | $1e5$ |    $1e6$ |
>
> **[Formatting]**
>
> Thanks for the suggestions on formatting. We have updated the main manuscript with the suggested changes.
>
> ---
> We would be glad to have further discussions if need be. Finally, we would appreciate it if our arguments change your mind towards supporting our work even more by increasing your score.
>
>
> **[1]** Van Seijen, Harm et al. "The LoCA regret: A consistent metric to evaluate model-based behavior in reinforcement learning." In Advances in Neural Information Processing Systems, volume 33, pp. 6562–6572. URL https://proceedings.neurips.cc/paper/2020/file/48db71587df6c7c442e5b76cc723169a-Paper.pdf
>
> **[2]** Wan, Yi et al. "Towards evaluating adaptivity of model-based reinforcement learning methods." In Proceedings of the 39th International Conference on Machine Learning, volume 162 of Proceedings of Machine Learning Research, pp. 22536–22561. URL https://proceedings.mlr.press/v162/wan22d.html

---

### Official Review · Reviewer_G6gQ · 2022-10-24

**Confidence:** 3
**Correctness:** 3
**Technical Novelty And Significance:** 3
**Empirical Novelty And Significance:** 3
**Recommendation:** 6

**Clarity, Quality, Novelty And Reproducibility:**

I found that the biggest quality issue of the paper is lacking proper empirical evaluations to demonstrate the usefulness of the approach in practice:
- In the LoCA setup, the only change between the two tasks (from Phase 1 to Phase 2) is the reward function. A possible (simple to implement, potentially strong) baseline would be to keep dynamics model (e.g., $P(s'|s, a)$) learned from Phase 1 and fine tune it in Phase 2, and train the reward prediction from scratch in Phase 2. This would allow all the data from the Phase 1 about the states to be still correct while discarding all the potentially incorrect reward information. On the flip side, I don't see any conceptual hurdles that prevent the proposed approach from working in the setup where both rewards and dynamics change (though I could be missing some details that left me with this wrong impression). Generalizing the LoCA setup to allow both dynamics and reward changes might make the paper more appealing as it solves a more general problem.
- In Figure 3, I found it to be a bit odd that the baselines used in the comparison are different across tasks. For example, the authors plotted against FIFO with only very large replay buffer sizes (3e6 and 4.5e6) in MountainCarLoCA, but used a wide range of replay buffer sizes in MiniGridLoCA (ranging from 2.5e4 to 1.8e6). Although the authors did mention that 4.5e6 worked the best for MountainCarLoCA, it is unclear what other replay buffer sizes have been attempted before for the MountainCarLoCA.
- It is possible that I am missing some detailed text, but I could not find the replay buffer size description for the Reacher experiments (for FIFO baseline). Was the best replay buffer size being reported in Figure 5? What replay buffer sizes have been considered?

**Strength And Weaknesses:**

*Strength*
- The proposed method is conceptually simple and can be combined with existing model-based RL methods
- The paper is well written with great clarity on the proposed method that seems reproducible.

*Weaknesses*
- I have doubts over the difficulties of the tasks tested in the paper. In particular, it is unclear to me that whether agents need to use the knowledge from the Phase 1 (environment with old reward function) to perform well in Phase 2 (environment with new reward function) at all. I cannot find the performance of training the agent from scratch directly in Phase 2 (e.g., in Figure 3 and Figure 5).
- The related work section is very terse and there are not many discussions how this work is similar or different from prior approaches in the continual learning literature (e.g., [1, 2, 3]). Some of them could be valid baselines. For example, instead of training the world model on the full replay buffer, we can train on a small subset of the most informative data from the Phase 1 environment (e.g., coreset [2]), or even discard the data from the Phase 1 environment completely and fine-tune the world model with parameter regularization on the data from the new environment only (e.g., EWC [1]).
- In the last paragraph, the authors wrote that "this is the first time–to the best of our knowledge that adaptivity for the deep MBRL methods has been shown." -- I think this statement is incorrect (see [4])

[1] Kirkpatrick, James, et al. "Overcoming catastrophic forgetting in neural networks." Proceedings of the national academy of sciences 114.13 (2017): 3521-3526.
[2] Nguyen, Cuong V., et al. "Variational continual learning." arXiv preprint arXiv:1710.10628 (2017).
[3] Rolnick, David, et al. "Experience replay for continual learning." Advances in Neural Information Processing Systems 32 (2019).
[4] Nagabandi, Anusha, Chelsea Finn, and Sergey Levine. "Deep online learning via meta-learning: Continual adaptation for model-based RL." arXiv preprint arXiv:1812.07671 (2018).

**Summary Of The Paper:**

The paper proposes a simple method for adaptation of model-based RL agent to new environments. Starting from the replay buffer filled with transitions from the old environment, the method works by gradually adding in the new transition from the new environment while removing data that are close to the states in the new transitions. In this way, the old data near the newly explored states in the new environment would not negatively hurt the learning of the world model while the old experience far away could still provide a good prior for the world model in the unknown states to prevent catastrophic forgetting. The closeness between two states required in this method is obtained by contrastive learning on a set of transitions collected by a random behavior policy in the new environment. The paper shows that the proposed method can improve upon the standard first-in-first-out (FIFO) replay buffer of different size on top of various existing model-based RL approaches in multiple custom tasks.

**Summary Of The Review:**

While the proposed adaptation approach is conceptually simple and the idea of local forgetting is appealing, it is unclear to me if the proposed approach is practically useful since the paper is missing the baseline of simply retraining a new RL agent from scratch in the new environment (Phase 2). In addition, the paper is missing important discussions and comparisons of the proposed method in relation to existing continual methods (which could serve as potentially strong baselines). Therefore, I would not recommend for acceptance of the paper at its current state.

A set of more thorough experiments with harder tasks where Phase 2 is difficult to solve without the experience of Phase 1 could potentially make the paper much stronger.

---

> ### Author Response · Authors · 2022-11-12
> **Response to Reviewer G6gQ (Part 1)**
>
>
> Thank you very much for this detailed review. We seem to have failed to properly explain our problem setting, as various of your comments and suggestions point to a key misunderstanding. We will further clarify our considered setting below and explain why some of the suggested baselines are not applicable and/or doomed to fail. We will also update the setting description and related work section in the paper to address these misunderstandings.
>
> **[Clarification of LoCA Setting]**
>
> *Re: "it is unclear to me that whether agents need to use the knowledge from the Phase 1 (environment with old reward function) to perform well in Phase 2 (environment with new reward function) at all." / "The only change between the two tasks (from Phase 1 to Phase 2) is the reward function"*
>
> Phase 1 knowledge is critical (by design); without it, no algorithm can perform well in phase 2. The reason for this can be observed in Figure 1: for training in phase 2 the initial state is drawn from a local region around T1, the T1-zone. And because the boundary of the T1-zone acts like a one-way passage, the agent cannot move out of the T1-zone once it's in there. Hence, during phase 2, the agent *only observes samples from within the T1-zone*. However, for evaluation in phase 2, the initial state is drawn *from the full state-space*. Hence, in order for an agent to perform well in Phase 2, it needs to know the optimal policy across the full state-space, even though this optimal policy changes radically between phase 1 and phase 2 and it only observes samples from the T1-zone in Phase 2. In Phase 1, the agent is trained using the full state-space. Hence, without re-using knowledge from Phase 1, the agent has no information about anything outside the T1-zone and is doomed to fail.
>
> More generally, to perform well in the LoCA setting, an agent needs to 1) learn a complete world model in phase 1 that is accurate throughout the full state-space, 2) in phase 2, update the world model for the T1-zone, while maintaining an accurate world model outside of the T1-zone, and 3) have a planning procedure that can compute a new optimal policy for the full state-space, in response to the local changes in the world model.
>
> Jointly satisfying these in the case of deep world models is challenging due to an interplay between catastrophic forgetting and interference. And our main contribution is that we show, for the first time, an algorithm that can achieve this.
>
> **[Missing Baselines]**
>
> *Re: Similarities with continual learning methods are not well explained, nor empirically compared against. For example, training the world model on only a small subset of the Phase 1 data could be a valid baseline,  or even to discard the Phase 1 data completely and only fine-tune the world model with parameter regularization (eg. EWC).*
>
> We would like to highlight that while our setting looks similar to continual learning, there are clear differences that make CL methods like EWC not suitable. In continual learning, the focus is not just to learn the current task better but also to not forget the previous task. The LoCA setup does not care about the previous task, and in fact, an ideal adaptive model-based RL agent would not do well in the previous task since it has updated its model according to the new local observations. Also, methods like EWC require the model to know when the task changed, while the LoCA setup does not reveal this information to the agent. Hence EWC is not applicable to this problem. We will highlight these connections and differences to CL in the related work section.
>
> As for empirical comparisons, in previous work, it has been shown already that discarding phase 1 samples, but keeping the world model and fine-tuning it in phase 2 does not work (Section 4, Figure 3, and Appendix B in [2]). The reason is that catastrophic forgetting causes the deep world model to become inaccurate outside of the T1-zone in phase 2.
>
>  >A possible (strong) baseline" is to keep the dynamics model and learn the reward function from scratch.
>
> We would like to highlight that this baseline is not possible since the agent does not get any signal when the reward function changes.

---

> > ### Author Response · Authors · 2022-11-12
> > **Response to Reviewer G6gQ (Part 2)**
> >
> > **[Novelty Claim]**
> >
> > *Re: The statement "this is the first time—to the best of our knowledge—that adaptivity for the deep MBRL methods has been shown."  is incorrect. See [4].*
> >
> > We are referring to a particular form of adaptivity, adaptivity to local changes in the environment, which is different from the adaptivity considered in [4]. We agree that the current statement is too general and will specify more clearly the type of adaptivity that we are referring to.
> >
> > The method in [4] is a meta-learning method, based on MAML. It requires meta-training tasks to train the prior on, which is not available in our setting. Furthermore, it requires observing several full trajectories from the new domain, to achieve faster adaptation to this domain relative to baselines. By contrast, our setting does not permit full trajectories from the new domain; full adaptation is required (i.e., optimal performance across the entire state-space) after only observing local information.
> >
> > **[Significance]**
> >
> > *Re: A set of more thorough experiments with harder tasks where Phase 2 is difficult to solve without the experience of Phase 1 could potentially make the paper much stronger.*
> >
> > First, as we explained above, in our current setting, it’s not possible to solve Phase 2 without experience from Phase 1. That being said, showing local adaptivity on harder tasks is one of our main goals for future work. That being said, we believe this work represents an important first step in that direction, as also acknowledged by Reviewer ‘oofa’. Note that in previous work [1,2], it has been shown that popular deep MBRL methods like MuZero and Dreamer fail to achieve effective adaptation in simple tasks like these.
> >
> > **[Replay Buffer size choices]**
> >
> > *Re: “In Figure 3, I found it to be a bit odd that the baselines used in the comparison are different across tasks. For example, the authors plotted against FIFO with only very large replay buffer sizes (3e6 and 4.5e6) in MountainCarLoCA, but used a wide range of replay buffer sizes in MiniGridLoCA.”*
> >
> > The reason is that evaluation with different buffer sizes for MountainCarLoCa has already been done in previous work; here, we simply use the sizes that were reported to work best. MiniGridLoCA is a new domain we introduce, and hence we used a wide range of replay buffer sizes.
> >
> > ---
> > If there are any further questions, we would be pleased to respond. We would appreciate it if our clarifications change your mind toward supporting our work by raising your score.
> >
> > **[1]** Van Seijen, Harm et al. "The LoCA regret: A consistent metric to evaluate model-based behavior in reinforcement learning." In Advances in Neural Information Processing Systems, volume 33, pp. 6562–6572. URL https://proceedings.neurips.cc/paper/2020/file/48db71587df6c7c442e5b76cc723169a-Paper.pdf
> >
> > **[2]** Wan, Yi et al. "Towards evaluating adaptivity of model-based reinforcement learning methods." In Proceedings of the 39th International Conference on Machine Learning, volume 162 of Proceedings of Machine Learning Research, pp. 22536–22561. URL https://proceedings.mlr.press/v162/wan22d.html

---

> > > ### Comment · Reviewer_G6gQ · 2022-12-12
> > > **Response to Author**
> > >
> > > Thanks for clarifying the problem setting! Somehow when I first read the paper I did not realize that the agent can only observe examples in the T1-zone in phase 2, which seemed to have caused a lot of misunderstandings on my side.
> > >
> > > My biggest remaining concern right now is that the experimental setup is a bit far from practical problems. (what practical problems approximately satisfy the T1-zone assumption?) It would be nice to show how useful the local forgetting replay buffer is in a more challenging setting where T1-zone in the future work.
> > >
> > > Despite this concern, I do think that the idea is nice and simple, and the empirical evaluations do show the effectiveness of the proposed approach. For this reason, I would like to raise my score to 6.

---

> > > > ### Author Response · Authors · 2022-12-12
> > > > **Response to Reviewer G6gQ**
> > > >
> > > > We really appreciate your comment and are delighted to see our clarification has changed your view on our submission. Below we address your remaining concern about the relation between the LoCA setup and practical problems.
> > > >
> > > > The LoCA setup is inspired by experimental setups used in neuroscience that are designed to determine whether a rodent exhibits model-free or model-based behavior (see first line Section 2). As such, the primary goal of the LoCA setup is not to model some common real-world problem scenario that the agent may encounter; instead, the setup is designed to accurately and robustly measure a behavioral feature that indicates model-based learning occurs. Similarly, the elaborate experimental setups that rodents are subjected to in neuroscience experiments do not reflect the typical situations these rodents may encounter outside the context of these experiments; their value lies in their ability to accurately measure a behavioral feature.
> > > >
> > > > The feature that is measured with the LoCA setup is effective adaptation to local changes in the environment, where 'effective' refers to the ability to compute the optimal policy under the changed environment in parts of the state-space that are not recently visited. This ability is important as it can be regarded as a 'behavioral signature' that indicates model-based learning. By contrast, a commonly used but poor indicator of model-based behavior is single-task performance, as there are many confounding factors that also improve single-task performance without requiring model-based learning (e.g. representation learning).
> > > >
> > > > We hope that this clarifies the significance of the LoCA setup. We will make further edits to the paper to better highlight this point.

---

### Official Review · Reviewer_54Vu · 2022-10-25

**Confidence:** 3
**Correctness:** 3
**Technical Novelty And Significance:** 2
**Empirical Novelty And Significance:** 2
**Recommendation:** 5

**Clarity, Quality, Novelty And Reproducibility:**

The paper is well-written and easy to follow. Based on the details provided in Appendix, I think the results can be reproduced with reasonable effort.

**Strength And Weaknesses:**

### Strength
The proposed method is easy to understand and efficient. Experiments showed expected results.

### Weakness
- Major
    I doubt whether the Local Change Adaptation (LoCA) is a suitable testbed for the proposed purpose. In phase 2, the agent is trapped around the T1 area, thus bringing a sharp distribution shift in the collected data during this phase. Data collected during this phase have a very poor state coverage, which hurts the world model learning severely. This designed small local change in the environment results in a huge shift in data distribution, which is a significant effect for MBRL that learns a policy during imagination. A more realistic case would be the case where only the reward function changes, while the agent movement not being limited in any way.
- Minor
    - Could the authors explain or provide experiment results analyzing the sensitivity of the hyper-param, $D_{local}$ and $N_{local}$ used in the experiments?

**Summary Of The Paper:**

This paper proposed a replay buffer strategy to help model-based RL models effectively adapt to local changes in the environment. When local changes happen, the traditional first-in-first-out (FIFO) data buffer will interfere with the model training due to the out-of-data samples still being stored in the buffer. To fix this, the authors proposed to update the buffer by only replacing the data in the local neighborhood of the newly observed samples. Contrastive learning method is used to learn the representation of states and to force consecutive states to have smaller Euclidean distance, thus neighborhood can be determined by applying a threshold on the distance measure.

**Summary Of The Review:**

This paper proposed a replay buffer strategy to help model-based RL models effectively adapt to local changes in the environment. However, the extreme cases designed in the experiments can not provide a general use for the proposed method.

---

> ### Author Response · Authors · 2022-11-12
> **Response to Reviewer 54Vu**
>
> We sincerely appreciate your comments on our work and value your concerns. We are pleased to see that you find our method easy to understand and efficient. The followings are our clarifications to your concerns:
>
> **[LoCA Setup]**
>
> *Re: “I doubt whether the Local Change Adaptation (LoCA) is a suitable testbed for the proposed purpose. In phase 2, the agent is trapped around the T1 area, thus bringing a sharp distribution shift in the collected data during this phase. Data collected during this phase have a very poor state coverage, which hurts the world model learning severely.”*
>
> First, the LoCA setup is not something we introduce; it was established by prior work published in two top-tier conferences (NeurIPS’20 [1] and ICML’22 [2]), demonstrating that (part of) the community definitely considers this setup relevant.  The motivation behind this setup is to have a reliable test that distinguishes model-based from model-free behavior, and it is inspired by tests used in neuroscience to distinguish model-based from model-free behavior in rodents and humans.
>
> A key characteristic that distinguishes model-based from model-free behavior is that model-based learning can propagate information effectively to parts of the state-space that have not recently been visited. The LoCA setup is designed to test this characteristic and the shift in data distribution is essential for this. Simply looking at the speed of adaptation without enforcing a shift in data distribution would make the test susceptible to various confounding factors that affect the speed of learning as well, such as representation learning.
>
> To be clear, there are other forms of adaptivity, as well as other behavioral characteristics, that are important too (i.e., exploration behavior) and not measured by the LoCA setup. As such, the setup is not meant to be a complete test of the efficacy of a method. Instead, it is meant to keep research on model-based learning on track, or at least, make people aware that many of the current deep MBRL methods don't exhibit a key feature of model-based behavior and encourage researchers to address this. Our submission takes an important step in this direction by demonstrating, for the first time, that a deep MBRL method can overcome the LoCA setup challenge.
>
>
> **[Sensitivity of the $D_{local}$ and $N_{local}$]**
>
> *Re: “Could the authors explain or provide experiment results analyzing the sensitivity of the hyper-param, $D_{local}$ and $N_{local}$ used in the experiments?”*
>
> We have provided Figure 8 and Figure 11 in the appendix of the main manuscript to show the sensitivity of the LoFo buffer to the $N_{local}$ hyperparameter. Based on these figures, we could say that our proposed method is not sensitive to the $N_{local}$. We are also running similar experiments for the $D_{local}$ hyperparameter and would add the respective figures besides those two.
>
> ---
> Please let us know if our clarifications successfully answer your concerns. We would be glad to have further discussions if need be. Finally, we would appreciate it if our arguments change your mind towards supporting our work even more by increasing your score.
>
>
> **[1]** Van Seijen, Harm et al. "The LoCA regret: A consistent metric to evaluate model-based behavior in reinforcement learning." In Advances in Neural Information Processing Systems, volume 33, pp. 6562–6572. URL https://proceedings.neurips.cc/paper/2020/file/48db71587df6c7c442e5b76cc723169a-Paper.pdf
>
> **[2]** Wan, Yi et al. "Towards evaluating adaptivity of model-based reinforcement learning methods." In Proceedings of the 39th International Conference on Machine Learning, volume 162 of Proceedings of Machine Learning Research, pp. 22536–22561. URL https://proceedings.mlr.press/v162/wan22d.html

---

### Author Response · Authors · 2022-11-18
**Follow Up**

Dear Reviewers, AC, and PC,

We want to express our gratitude to the reviewers once more for their time and effort. We are aware that reviewing comes with a huge workload. However, since the discussion period is about to end soon, we wanted to kindly remind the reviewers to go through our responses.

While all reviewers agreed on the fact that our proposed solution to overcome the LoCA setup challenge is simple and easy to understand, various suggestions made by the "G6gQ" reviewer pointed to some key misunderstandings regarding the LoCA setup. We tried to clarify the questions of the "G6gQ" and "54Vu" reviwers on the LoCA setup in our responses. And, with the time given, we would be happy to provide further clarifications if need be.

---

### Author Response · Authors · 2022-12-09
**Follow Up and Reminder**

Dear AC, PC, and reviewers,

We want to express our gratitude to the reviewers once more for their time and feedback. We hope our response has clarified some of the key misunderstandings and answered all the reviewer's questions and concerns. Since we have not received any response to our rebuttal yet, we wanted to send a small reminder. We are happy to answer any further questions that reviewers might have.

Best,

The authors

---

### Decision · Program_Chairs · 2023-01-20

**Decision:**

Reject

**Justification For Why Not Higher Score:**

Proposed technique seems engineered for a somewhat restrictive set of evaluation environments. Needs more extensive evaluation to justify broader utility.

**Justification For Why Not Lower Score:**

N/A

**Metareview: Summary, Strengths And Weaknesses:**

The paper proposes a replay buffer update strategy for adaptive RL. The strategy is conceptually simple and makes sense: instead of adopting a FIFO update rule to the replay buffer, the authors propose to replace samples closest to the newly observed samples based on some measure of similarity. Empirically, the paper demonstrates benefits on the LoCo environments proposed in recent work.

On the positive side, the reviewers appreciated the clarity of the work and the fact that the proposed method is conceptually simple and achieves the expected results. However, there were uniform concerns among the reviewers on the significance of the LoCo environments for more practical settings and consequently, how engineered is the proposed update rule for these environments. To this end, the paper could be improved on two major fronts: (a) experiments on conventional benchmarks without the LoCo variant --- the proposed strategy is presented as a plug-and-play since it only modifies the update protocols for the replay buffer, so it should be relatively easy to test (b) more baselines for updating replay buffers, e.g., prioritized/attentive experience replay --- the authors state these techniques as complementary, but without experimental evidence, their combined effects are far from guaranteed to add up. As it stands, the LoCo environments should inspire model-based RL techniques that can transfer to realistic settings rather than ones that can solve only the LoCo environments in isolation. With the above set of experiments, we expect to better understand the significance of the proposed buffer update rule in more realistic settings.  I encourage the authors to follow up on these suggestions for a stronger future version.